# COULOMB GANS: PROVABLY OPTIMAL NASH EQUILIBRIA VIA POTENTIAL FIELDS

**Thomas Unterthiner**[1]    **Bernhard Nessler**[1]    **Calvin Seward**[1,2]    **Günter Klambauer**[1]

**Martin Heusel**[1]                **Hubert Ramsauer**[1]                **Sepp Hochreiter**[1]

[1]LIT AI Lab & Institute of Bioinformatics, Johannes Kepler University Linz, Austria
[2]Zalando Research, Mühlenstraße 25, 10243 Berlin, Germany
`{unterthiner,nessler,seward,klambauer,mhe,ramsauer,hochreit}@bioinf.jku.at`

## ABSTRACT

Generative adversarial networks (GANs) evolved into one of the most successful unsupervised techniques for generating realistic images. Even though it has recently been shown that GAN training converges, GAN models often end up in local Nash equilibria that are associated with mode collapse or otherwise fail to model the target distribution. We introduce Coulomb GANs, which pose the GAN learning problem as a potential field, where generated samples are attracted to training set samples but repel each other. The discriminator learns a potential field while the generator decreases the energy by moving its samples along the vector (force) field determined by the gradient of the potential field. Through decreasing the energy, the GAN model learns to generate samples according to the whole target distribution and does not only cover some of its modes. We prove that Coulomb GANs possess only one Nash equilibrium which is optimal in the sense that the model distribution equals the target distribution. We show the efficacy of Coulomb GANs on LSUN bedrooms, CelebA faces, CIFAR-10 and the Google Billion Word text generation.

## 1 INTRODUCTION

Generative adversarial networks (GANs) (Goodfellow et al., 2014) excel at constructing realistic images (Radford et al., 2016; Ledig et al., 2016; Isola et al., 2017; Arjovsky et al., 2017; Berthelot et al., 2017) and text (Gulrajani et al., 2017). In GAN learning, a discriminator network guides the learning of another, generative network. This procedure can be considered as a game between the generator which constructs synthetic data and the discriminator which separates synthetic data from training set data (Goodfellow, 2017). The generator's goal is to construct data which the discriminator cannot tell apart from training set data. GAN convergence points are local Nash equilibria. At these local Nash equilibria neither the discriminator nor the generator can locally improve its objective.

Despite their recent successes, GANs have several problems. First (I), until recently it was not clear if in general gradient-based GAN learning could converge to one of the local Nash equilibria (Salimans et al., 2016; Goodfellow, 2014; Goodfellow et al., 2014). It is even possible to construct counterexamples (Goodfellow, 2017). Second (II), GANs suffer from "mode collapsing", where the model generates samples only in certain regions which are called modes. While these modes contain realistic samples, the variety is low and only a few prototypes are generated. Mode collapsing is less likely if the generator is trained with batch normalization, since the network is bound to create a certain variance among its generated samples within one batch (Radford et al., 2016; Chintala et al., 2016). However batch normalization introduces fluctuations of normalizing constants which can be harmful (Klambauer et al., 2017; Goodfellow, 2017). To avoid mode collapsing without batch normalization, several methods have been proposed (Che et al., 2017; Metz et al., 2016; Salimans et al.,

2016). Third (III), GANs cannot assure that the density of training samples is correctly modeled by the generator. The discriminator only tells the generator whether a region is more likely to contain samples from the training set or synthetic samples. Therefore the discriminator can only distinguish the support of the model distribution from the support of the target distribution. Beyond matching the support of distributions, GANs with proper objectives may learn to locally align model and target densities via averaging over many training examples. On a global scale, however, GANs fail to equalize model and target densities. The discriminator does not inform the generator globally where probability mass is missing. Consequently, standard GANs are not assured to capture the global sample density and are prone to neglect large parts of the target distribution. The next paragraph gives an example of this. Fourth (IV), the discriminator of GANs may forget previous modeling errors of the generator which then may reappear, a property that leads to oscillatory behavior instead of convergence (Goodfellow, 2017).

Recently, problem (I) was solved by proving that GAN learning does indeed converge when discriminator and generator are learned using a two time-scale learning rule (Heusel et al., 2017). Convergence means that the expected SGD-gradient of both the discriminator objective and the generator objective are zero. Thus, neither the generator nor the discriminator can locally improve, i.e., learning has reached a local Nash equilibrium. However, convergence alone does not guarantee good generative performance. It is possible to converge to sub-optimal solutions which are local Nash equilibria. Mode collapse is a special case of a local Nash equilibrium associated with sub-optimal generative performance. For example, assume a two mode real world distribution where one mode contains too few and the other mode too many generator samples. If no real world samples are between these two distinct modes, then the discriminator penalizes to move generated samples outside the modes. Therefore the generated samples cannot be correctly distributed over the modes. Thus, standard GANs cannot capture the global sample density such that the resulting generators are prone to neglect large parts of the real world distribution. A more detailed example is listed in the Appendix in Section A.1.

In this paper, we introduce a novel GAN model, the Coulomb GAN, which has only one Nash equilibrium. We are later going to show that this Nash equilibrium is optimal, i.e., the model distribution matches the target distribution. We propose Coulomb GANs to avoid the GAN shortcoming (II) to (IV) by using a potential field created by point charges analogously to the electric field in physics. The next section will introduce the idea of learning in a potential field and prove that its only solution is optimal. We will then show how learning the discriminator and generator works in a Coulomb GAN and discuss the assumptions needed for our optimality proof. In Section 3 we will then see that the Coulomb GAN does indeed work well in practice and that the samples it produces have very large variability and appear to capture the original distribution very well.

**Related Work.** Several GAN approaches have been suggested for bringing the target and model distributions in alignment using not just local discriminator information: Geometric GANs combine samples via a linear support vector machine which uses the discriminator outputs as samples, therefore they are much more robust to mode collapsing (Lim & Ye, 2017). Energy-Based GANs (Zhao et al., 2017) and their later improvement BEGANs (Berthelot et al., 2017) optimize an energy landscape based on auto-encoders. McGANs match mean and covariance of synthetic and target data, therefore are more suited than standard GANs to approximate the target distribution (Mroueh et al., 2017). In a similar fashion, Generative Moment Matching Networks (Li et al., 2015) and MMD nets (Dziugaite et al., 2015) directly optimize a generator network to match a training distribution by using a loss function based on the maximum mean discrepancy (MMD) criterion (Gretton et al., 2012). These approaches were later expanded to include an MMD criterion with learnable kernels and discriminators (Li et al., 2017). The MMD criterion that these later approaches optimize has a form similar to the energy function that Coulomb GANs optimize (cf. Eq. (33)). However, all MMD approaches end up using either Gaussian or Laplace kernels, which are not guaranteed to find the optimal solution where the model distribution matches the target distribution. In contrast, the Plummer kernel which is employed in this work has been shown to lead to the optimal solution (Hochreiter & Obermayer, 2005). We show that even a simplified version of the Plummer kernel, the low-dimensional Plummer kernel, ensures that gradient descent convergences to the optimal solution as stated by Theorem 1. Furthermore, most MMD GAN approaches use the MMD directly as loss function though the number of possible samples in a mini-batch is limited. Therefore MMD approaches face a sampling problem in high-dimensional spaces. The Coulomb GAN instead learns a discriminator network that gradually improves its approximation of the potential field via learning

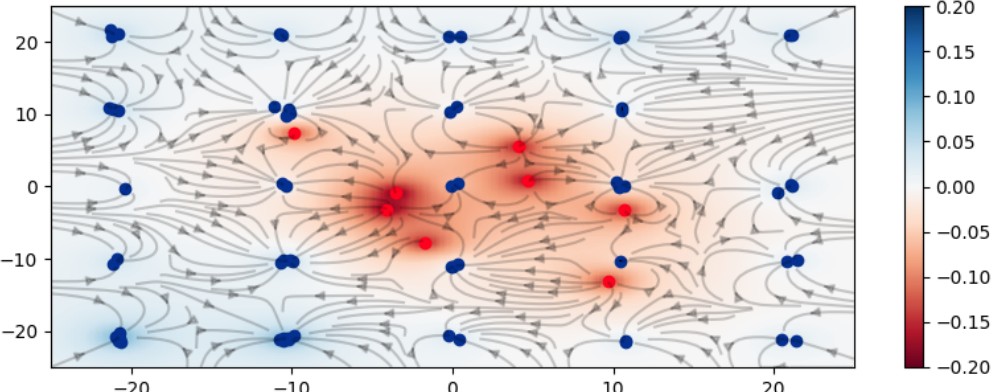

Figure 1: The vector field of a Coulomb GAN. The basic idea behind the Coulomb GAN: true samples (blue) and generated samples (red) create a potential field (scalar field). Blue samples act as sinks that attract the red samples, which repel each other. The superimposed vector field shows the forces acting on the generator samples to equalize potential differences, and the background color shows the potential at each position. Best viewed in color.

on many mini-batches. The discriminator network also tracks the slowly changing generator distribution during learning. Most importantly however, our approach is, to the best of our knowledge, the first one for which optimality, i.e., ability to perfectly learn a target distribution, can be proved.

The use of the Coulomb potential for learning is not new. Coulomb Potential Learning was proposed to store arbitrary many patterns in a potential field with perfect recall and without spurious patterns (Perrone & Cooper, 1995). Another related work is the Potential Support Vector Machine (PSVM), which minimizes Coulomb potential differences (Hochreiter & Mozer, 2001; Hochreiter et al., 2003). Hochreiter & Obermayer (2005) also used a potential function based on Plummer kernels for optimal unsupervised learning, on which we base our work on Coulomb GANs.

## 2 COULOMB GANS

### 2.1 GENERAL CONSIDERATIONS ON GANS

We assume data samples $\boldsymbol{a} \in \mathbb{R}^m$ for a model density $p_x(.)$ and a target density $p_y(.)$. The goal of GAN learning is to modify the model in a way to obtain $p_x(.) = p_y(.)$. We define the **difference of densities** $\rho(\boldsymbol{a}) = p_y(\boldsymbol{a}) - p_x(\boldsymbol{a})$ which should be pushed toward zero for all $\boldsymbol{a} \in \mathbb{R}^m$ during learning. In the GAN setting, the discriminator $D(\boldsymbol{a})$ is a function $D : \mathbb{R}^m \to \mathbb{R}$ that learns to discriminate between generated and target samples and predicts how likely it is that $\boldsymbol{a}$ is sampled from the target distribution. In conventional GANs, $D(\boldsymbol{a})$ is usually optimized to approximate the probability of seeing a target sample, or $\rho(\boldsymbol{a})$ or some similar function. The generator $G(\boldsymbol{z})$ is a continuous function $G : \mathbb{R}^n \to \mathbb{R}^m$ which maps some $n$-dimensional random variable $\boldsymbol{z}$ into the space of target samples. $\boldsymbol{z}$ is typically sampled from a multivariate Gaussian or Uniform distribution.

In order to improve the generator, a GAN uses the gradient of the discriminator $\boldsymbol{\nabla}_a D(\boldsymbol{a})$ with respect to the discriminator input $\boldsymbol{a} = G(\boldsymbol{z})$ for learning. The objective of the generator is a scalar function $D(G(\boldsymbol{z}))$, therefore the gradient of the objective function is just a scaled version of the gradient $\boldsymbol{\nabla}_a D(\boldsymbol{a})$ which would then propagate further to the parameters of $G$. This gradient $\boldsymbol{\nabla}_a D(\boldsymbol{a})$ tells the generator in which direction $\rho(\boldsymbol{a})$ becomes larger, i.e., in which direction the ratio of target examples increases. The generator changes slightly so that $\boldsymbol{z}$ is now mapped to a new $\boldsymbol{a}' = G'(\boldsymbol{z})$, moving the sample generated by $\boldsymbol{z}$ a little bit towards the direction where $\rho(\boldsymbol{a})$ was larger, i.e., where target examples were more likely. However, $\rho(\boldsymbol{a})$ and its derivative only take into account the local neighborhood of $\boldsymbol{a}$, since regions of the sample space that are distant from $\boldsymbol{a}$ do not have much influence on $\rho(\boldsymbol{a})$. Regions of data space that have strong support in $p_y$ but not in $p_x$ will not be noticed by the generator via discriminator gradients. The restriction to local environments hampers GAN learning significantly (Arjovsky & Bottou, 2017; Arjovsky et al., 2017).

The theoretical analysis of GAN learning can be done at three different levels: (1) in the space of distributions $p_x$ and $p_y$ regardless of the fact that $p_x$ is realized by $G$ and $p_z$, (2) in the space of functions $G$ and $D$ regardless of the fact that $G$ and $D$ are typically realized by a parametric form, i.e., as neural networks, or (3) in the space of the parameters of $G$ and $D$. Goodfellow et al. (2014) use (1) to prove convergence of GAN learning in their Proposition 2 in a hypothetical scenario where the learning algorithm operates by making small, local moves in $p_x$ space. In order to see that level (1) and (2) should both be understood as hypothetical scenarios, remember that in all practical implementations, $p_x$ can only be altered implicitly by making small changes to the generator function G, which in turn can only be changed implicitly by small steps in its parameters. Even if we assume that the mapping from a distribution $p_x$ to the generator $G$ that induced it exists and is unique, this mapping from $p_x$ to the space of $G$ is not continuous. To see this, consider changing a distribution $p1_x$ to a new distribution $p2_x$ by moving a small amount $\epsilon$ of its density to an isolated region in space where $p1_x$ has no support. Let's further assume this region has distance $d$ to any other regions of support of $p1_x$. By letting $\epsilon \to 0$, the distance between $p1_x$ and $p2_x$ becomes smaller, yet the distance between the inducing generator functions $G_1$ and $G_2$ (e.g. using the supremum norm on bounded functions) will not tend to zero because for at least one function input $\boldsymbol{z}$ we have: $|G_1(\boldsymbol{z}) - G_2(\boldsymbol{z})| \geqslant d$. Because of this, we need to go further than the distribution space when analyzing GAN learning. In practice, when learning GANs, we are restricted to small steps in parameter space, which in turn lead to small steps in function space and finally to small steps in distribution space. But not all small steps in distribution space can be realized this way as shown in the example above. This causes local Nash equilibria in the function space, because even though in distribution space it would be easy to escape by making small steps, such a step would require very large changes in function space and is thus not realizable. In this paper we show that Coulomb GANs do not exhibit any local Nash equilibria in the space of the functions $G$ and $D$. To the best of our knowledge, this is the first formulation of GAN learning that can guarantee this property. Of course, Coulomb GANs are learned as parametrized neural networks, and as we will discuss in Subsection 2.4.2, Coulomb GANs are not immune to the usual issues that arise from parameter learning, such as over- and underfitting, which can cause local Nash Equilibria due to a bad choice of parameters.

## 2.2 From Conventional GANs to Potentials

If the density $p_x(.)$ or $p_y(.)$ approaches a Dirac delta-distribution, gradients vanish since the density approaches zero except for the exact location of data points. Similarly, electric point charges are often represented by Dirac delta-distributions, however the electric potential created by a point charge has influence everywhere in the space, not just locally. The electric potential (Coulomb potential) created by the point charge $Q$ is $\Phi_C = \frac{1}{4\pi\varepsilon_0}\frac{Q}{r}$, where $r$ is the distance to the location of $Q$ and $\varepsilon_0$ is the dielectric constant. Motivated by this electric potential, we introduce a similar concept for GAN learning: Instead of the difference of densities $\rho(\boldsymbol{a})$, we rather consider a **potential function** $\Phi(\boldsymbol{a})$ defined as

$$\Phi(\boldsymbol{a}) = \int \rho(\boldsymbol{b})\, k(\boldsymbol{a}, \boldsymbol{b})\, \mathrm{d}\boldsymbol{b}\,, \tag{1}$$

with some kernel $k(\boldsymbol{a}, \boldsymbol{b})$ which defines the influence of a point at $\boldsymbol{b}$ onto a point at $\boldsymbol{a}$. The crucial advantage of potentials $\Phi(\boldsymbol{a})$ is that each point can influence each other point in space if $k$ is chosen properly. If we minimize this potential $\Phi(\boldsymbol{a})$ we are at the same time minimizing the difference of densities $\rho(\boldsymbol{a})$: For all kernels $k$ it holds that if $\rho(\boldsymbol{b}) = 0$ for all $\boldsymbol{b}$ then $\Phi(\boldsymbol{a}) = 0$ for all $\boldsymbol{a}$. We must still show that (i) $\Phi(\boldsymbol{a}) = 0$ for all $\boldsymbol{a}$ then $\rho(\boldsymbol{b}) = 0$ for all $\boldsymbol{b}$, and even more importantly, (ii) whether a gradient optimization of $\Phi(\boldsymbol{a})$ leads to $\Phi(\boldsymbol{a}) = 0$ for all $\boldsymbol{a}$. This is not the case for every kernel. Indeed only for particular kernels $k$ gradient optimization of $\Phi(\boldsymbol{a})$ leads to $\rho(\boldsymbol{b}) = 0$ for all $\boldsymbol{b}$, that is, $p_x(\boldsymbol{b}) = p_y(\boldsymbol{b})$ for all $\boldsymbol{b}$ (Hochreiter & Obermayer, 2005) (see also Theorem 1 below). An example for such a kernel $k$ is the one leading to the Coulomb potential $\Phi_C$ from above, where $k(\boldsymbol{a}, \boldsymbol{b}) = \frac{1}{\|\boldsymbol{a}-\boldsymbol{b}\|}$ for $m = 3$. As we will see in the following, the ability to have samples that influence each other over long distances, like charges in a Coulomb potential, will lead to GANs with a single, optimal Nash equilibrium.

### 2.3 GANs as Electrical Fields

For Coulomb GANs, the generator objective is derived from electrical field dynamics: real and generated samples generate a potential field, where samples of the same class (real vs. generated) repel each other, but attract samples of the opposite class. However, real data points are fixed in space, so the only samples that can move are the generated ones. In turn, the gradient of the potential with respect to the input samples creates a vector field in the space of samples. The generator can move its samples along the forces generated by this field. Such a field is depicted in Figure 1. The discriminator learns to predict the potential function, in order to approximate the current potential landscape of all samples, not just the ones in the current mini-batch. Meanwhile, the generator learns to distribute its samples across the whole field in such a way that the energy is minimized, thus naturally avoids mode collapse and covering the whole region of support of the data. The energy is minimal and equal to zero only if all potential differences are zero and the model distribution is equal to the target distribution.

Within an electrostatic field, the strength of the force on one particle depends on its distance to other particles and their charges. If left to move freely, the particles will organize themselves into a constellation where all forces equal out and no potential differences are present. For continuous charge distributions, the potential field is constant without potential differences if charges no longer move since forces are equaled out. If the potential field is constant, then the difference of densities $\rho$ is constant, too. Otherwise the potential field would have local bumps. The same behavior is modeled within our Coulomb GAN, except that real and generated samples replace the positive and negative particles, respectively, and that the real data points remain fixed. Only the generated samples are allowed to move freely, in order to minimize $\rho$. The generated samples are attracted by real samples, so they move towards them. At the same time, generated samples should repel each other, so they do not clump together, which would lead to mode collapsing.

Analogously to electrostatics, the potential $\Phi(\boldsymbol{a})$ from Eq. (1) gives rise to a **field** $\mathbf{E}(\boldsymbol{a}) = -\boldsymbol{\nabla}_a \Phi(\boldsymbol{a})$. and to an **energy function** $F(\rho) = \frac{1}{2} \int \rho(\boldsymbol{a}) \Phi(\boldsymbol{a}) \mathrm{d}\boldsymbol{a}$. The field $\mathbf{E}(\boldsymbol{a})$ applies a force on charges at $\boldsymbol{a}$ which pushes the charges toward lower energy constellations. Ultimately, the Coulomb GAN aims to make the potential $\Phi$ zero everywhere via the field $\mathbf{E}(\boldsymbol{a})$, which is the negative gradient of $\Phi$. For proper kernels $k$, it can be shown that (i) $\Phi$ can be pushed to zero via its negative gradient given by the field and (ii) that $\Phi(\boldsymbol{a}) = 0$ for all $\boldsymbol{a}$ implies $\rho(\boldsymbol{a}) = 0$ for all $\boldsymbol{a}$, therefore, $p_x(\boldsymbol{a}) = p_y(\boldsymbol{a})$ for all $\boldsymbol{a}$ (Hochreiter & Obermayer, 2005) (see also Theorem 1 below).

#### 2.3.1 Learning Process

During learning we do not change $\Phi$ or $\rho$ directly. Instead, the location $\boldsymbol{a} = G(\boldsymbol{z})$ to which the random variable $\boldsymbol{z}$ is mapped changes to a new location $\boldsymbol{a}' = G'(\boldsymbol{z})$. For the GAN optimization dynamics, we assume that generator samples $\boldsymbol{a} = G(\boldsymbol{z})$ can move freely, which is ensured by a sufficiently complex generator. Importantly, generator samples originating from random variables $\boldsymbol{z}$ do neither disappear nor are they newly created but are conserved. This conservation is expressed by the continuity equation (Schwartz, 1972) that describes how the difference between distributions $\rho(\boldsymbol{a})$ changes as the particles are moving along the field, i.e., how moving samples during the learning process changes our densities:

$$\dot{\rho}(\boldsymbol{a}) = -\boldsymbol{\nabla} \cdot (\rho(\boldsymbol{a}) \, \boldsymbol{v}(\boldsymbol{a})) \tag{2}$$

for sample density difference $\rho$ and unit charges that move with "velocity" $\boldsymbol{v}(\boldsymbol{a}) = \mathrm{sign}(\rho(\boldsymbol{a}))\mathbf{E}(\boldsymbol{a})$. The continuity equation is crucial as it establishes the connection between moving samples and changing the generator density and thereby $\rho$. The sign function of the velocity indicates whether positive or negative charges are present at $\boldsymbol{a}$. The divergence operator "$\boldsymbol{\nabla}\cdot$" determines whether samples move toward or outward of $\boldsymbol{a}$ for a given field. Basically, the continuity equation says that if the generator density increases, then generator samples must flow into the region and if the generator density decreases, they flow outwards. We assume that differently charged particles cancel each other. If generator samples are moved away from a location $\boldsymbol{a}$ then $\rho(\boldsymbol{a})$ is increasing while $\rho(\boldsymbol{a})$ is decreasing when generator samples are moved toward $\boldsymbol{a}$. The continuity equation is also obtained as a first order ODE to move particles in a potential field (Dembo & Zeitouni, 1988), therefore describes the dynamics how the densities are changing. We obtain

$$\dot{\rho}(\boldsymbol{a}) = -\mathrm{sign}(\rho(\boldsymbol{a})) \, \boldsymbol{\nabla} \cdot (\rho(\boldsymbol{a}) \, \mathbf{E}(\boldsymbol{a})) = -\boldsymbol{\nabla} \cdot (|\rho(\boldsymbol{a})| \, \mathbf{E}(\boldsymbol{a})) . \tag{3}$$

The density difference $\rho(\boldsymbol{a})$ indicates how many samples are locally available for being moved. At each local minimum and local maximum $\boldsymbol{a}$ of $\rho$ we obtain $\boldsymbol{\nabla}_a \rho(\boldsymbol{a}) = 0$. Using the product rule for the divergence operator, at points $\boldsymbol{a}$ that are minima or maxima, Eq. (3) reduces to

$$\dot{\rho}(\boldsymbol{a}) = -\operatorname{sign}(\rho(\boldsymbol{a})) \, \rho(\boldsymbol{a}) \, \boldsymbol{\nabla} \cdot \mathbf{E}(\boldsymbol{a}) \, . \tag{4}$$

In order to ensure that $\rho$ converges to zero, it is necessary and sufficient that $\operatorname{sign}(\boldsymbol{\nabla} \cdot \mathbf{E}(\boldsymbol{a})) = \operatorname{sign}(\rho(\boldsymbol{a}))$, where $\div_a \rho(\boldsymbol{a}) = 0$, as this condition ensures the uniform decrease of the maximal absolute density differences $|\rho(\boldsymbol{a}_{\max})|$.

### 2.3.2 CHOICE OF KERNEL

As discussed before, the choice of kernel is crucial for Coulomb GANs. The $m$-dimensional Coulomb kernel and the $m$-dimensional Plummer kernel lead to (i) $\Phi$ that is pushed to zero via the field it creates and (ii) that $\Phi(\boldsymbol{a}) = 0$ for all $\boldsymbol{a}$ implies $\rho(\boldsymbol{a}) = 0$ for all $\boldsymbol{a}$, therefore, $p_x(\boldsymbol{a}) = p_y(\boldsymbol{a})$ for all $\boldsymbol{a}$ (Hochreiter & Obermayer, 2005). Thus, gradient learning with these kernels has been proved to converge to an optimal solution. However, both the $m$-dimensional Coulomb and the $m$-dimensional Plummer kernel lead to numerical instabilities if $m$ is large. Therefore the Coulomb potential $\Phi(\boldsymbol{a})$ for the Coulomb GAN was constructed by a low-dimensional Plummer kernel $k$ with parameters $d \leqslant m - 2$ and $\epsilon$:

$$\Phi(\boldsymbol{a}) = \int \rho(\boldsymbol{b}) \, k(\boldsymbol{a}, \boldsymbol{b}) \, \mathrm{d}\boldsymbol{b} \, , \quad k(\boldsymbol{a}, \boldsymbol{b}) = \frac{1}{(\sqrt{\|\boldsymbol{a} - \boldsymbol{b}\|^2 + \epsilon^2})^d} \, . \tag{5}$$

The original Plummer kernel is obtained with $d = m - 2$. The resulting field and potential energy is

$$\mathbf{E}(\boldsymbol{a}) = -\int \rho(\boldsymbol{b}) \, \boldsymbol{\nabla}_a k(\boldsymbol{a}, \boldsymbol{b}) \, \mathrm{d}\boldsymbol{b} = -\boldsymbol{\nabla}_a \, \Phi(\boldsymbol{a}) \, , \tag{6}$$

$$F(\rho) = \frac{1}{2} \int \rho(\boldsymbol{a}) \, \Phi(\boldsymbol{a}) \, \mathrm{d}\boldsymbol{a} = \frac{1}{2} \int \int \rho(\boldsymbol{a}) \, \rho(\boldsymbol{b}) \, k(\boldsymbol{a}, \boldsymbol{b}) \, \mathrm{d}\boldsymbol{b} \, \mathrm{d}\boldsymbol{a} \, . \tag{7}$$

The next theorem states that for freely moving generated samples, $\rho$ converges to zero, that is, $p_x(.) = p_y(.)$, when using this potential function $\Phi(\boldsymbol{a})$.

**Theorem 1** (Convergence with low-dimensional Plummer kernel). *For $\boldsymbol{a}, \boldsymbol{b} \in \mathbb{R}^m$, $d \leqslant m - 2$, and $\epsilon > 0$ the densities $p_x(.)$ and $p_y(.)$ equalize over time when minimizing energy $F$ with the low-dimensional Plummer kernel by gradient descent. The convergence is faster for larger $d$.*

*Proof.* See Section A.2. $\qquad\square$

### 2.4 DEFINITION OF THE COULOMB GAN

The Coulomb GAN minimizes the electric potential energy from Eq. (6) using a stochastic gradient descent based approach using mini-batches. Appendix Section A.4 contains the equations for the Coulomb potential, field, and energy in this case. Generator samples are obtained by drawing $N_x$ random numbers $\boldsymbol{z}_i$ and transforming them into outputs $\boldsymbol{x}_i = G(\boldsymbol{z}_i)$. Each mini-batch also includes $N_y$ real world samples $\boldsymbol{y}_i$. This gives rise to a mini-batch specific potential, where in Eq. (5) we use $\rho(\boldsymbol{a}) = p_y(\boldsymbol{a}) - p_x(\boldsymbol{a})$ and replace the expectations by empirical means using the drawn samples:

$$\hat{\Phi}(\boldsymbol{a}) = \frac{1}{N_y} \sum_{i=1}^{N_y} k(\boldsymbol{a}, \boldsymbol{y}_i) - \frac{1}{N_x} \sum_{i=1}^{N_x} k(\boldsymbol{a}, \boldsymbol{x}_i) \, . \tag{8}$$

It is tempting to have a generator network that directly minimizes this potential $\hat{\Phi}$ between generated and training set points. In fact, we show that $\hat{\Phi}$ is an unbiased estimate for $\Phi$ in Appendix Section A.4. However, the estimate has very high variance: for example, if a mini-batch fails to sample training data from an existing mode, the field would drive all generated samples that have been generated at this mode to move elsewhere. The high variance has to be counteracted by extremely low learning rates, which makes learning infeasible in practice, as confirmed by initial experiments. Our solution to this problem is to have a network that generalizes over the mini-batch specific potentials: each mini-batch contains different generator samples $\mathcal{X} = \boldsymbol{x}_i$ for $i = 1, \ldots, N_x$ and real world samples $\mathcal{Y} = \boldsymbol{y}_i$ for $i = 1, \ldots, N_y$, they create a batch-specific potential $\hat{\Phi}$. The goal of the

discriminator is to learn $\mathrm{E}_{\mathcal{X},\mathcal{Y}}(\hat{\Phi}(\boldsymbol{a})) = \Phi(\boldsymbol{a})$, i.e., the potential averaged over many mini-batches. Thus the discriminator function $D$ fulfills a similar role as other typical GAN discriminator functions, i.e., it discriminates between real and generated data such that for any point in space $\boldsymbol{a}$, $D(\boldsymbol{a})$ should be greater than zero if the $p_y(\boldsymbol{a}) > p_x(\boldsymbol{a})$ and smaller than zero otherwise. In particular $D(\boldsymbol{a})$ also indicates, via its gradient and its potential properties, directions toward regions where training set samples are predominant and where generator samples are predominant.

The generator in turn tries to move all of its samples according to the vector field into areas where generator samples are missing and training set samples are predominant. The generator minimizes the approximated energy $F$ as predicted by the discriminator. The loss $\mathcal{L}_D$ for the discriminator and $\mathcal{L}_G$ for the generator are given by:

$$\mathcal{L}_D(D;G) = \frac{1}{2}\,\mathrm{E}_{p_a}\left((D(\boldsymbol{a}) - \hat{\Phi}(\boldsymbol{a}))^2\right) \tag{9}$$

$$\mathcal{L}_G(G;D) = -\frac{1}{2}\,\mathrm{E}_{p_z}\left(D(G(\boldsymbol{z}))\right) . \tag{10}$$

Where $p(\boldsymbol{a}) = 1/2\int \mathcal{N}(\boldsymbol{a};G(\boldsymbol{z}),\epsilon\boldsymbol{I})p_z(\boldsymbol{z})\mathrm{d}\boldsymbol{z} + 1/2\int\mathcal{N}(\boldsymbol{a};\boldsymbol{y},\epsilon\boldsymbol{I})p_y(\boldsymbol{y})\mathrm{d}\boldsymbol{y}$, i.e., a distribution where each point of support both of the generator and the real world distribution is surrounded with a Gaussian ball of width $\epsilon\boldsymbol{I}$ similar to Bishop et al. (1998), in order to overcome the problem that the generator distribution is only a sub-manifold of $\mathbb{R}^m$. These loss functions cause the approximated potential values $D(\boldsymbol{a})$ that are negative are pushed toward zero. Finally, the Coulomb GAN, like all other GANs, consists of two parts: a generator to generate model samples, and a discriminator that provides its learning signal. Without a discriminator, our would be very similar to GMMNs (Li et al., 2015), as can be seen in Eq. (33), but with an optimal Kernel specifically tailored to the problem of estimating differences between probability distributions.

We use each mini-batch only for one update of the discriminator and the generator. It is important to note that the discriminator uses each sample in the mini batch twice: once as a point to generate the mini-batch specific potential $\hat{\Phi}$, and once as a point in space for the evaluation of the potential $\hat{\Phi}$ and its approximation $D$. Using each sample twice is done for performance reasons, but not strictly necessary: the discriminator could learn the potential field by sampling points that lie between generator and real samples as in Gulrajani et al. (2017), but we are mainly interested in correct predictions in the vicinity of generator samples. Pseudocode for the learning algorithm is detailed in Algorithm 1 in the appendix.

### 2.4.1 OPTIMALITY OF THE SOLUTION

Convergence of the GAN learning process was proved for a two time-scales update rule by Heusel et al. (2017). A local Nash equilibrium is a pair of generator and discriminator $(D^*, G^*)$ that fulfills the two conditions

$$D^* = \underset{D\in U(D^*)}{\arg\min}\,\mathcal{L}_D(D;G^*) \quad \text{and} \quad G^* = \underset{G\in U(G^*)}{\arg\min}\,\mathcal{L}_G(G;D^*) .$$

for some neighborhoods $U(D^*)$ and $U(G^*)$. We show in the following Theorem 2 that for Coulomb GANs every local Nash equilibrium necessarily is identical to the unique global Nash equilibrium. In other words, any equilibrium point of the Coulomb GAN that is found to be local optimal has to be the one global Nash equilibrium as the minimization of the energy $F(\rho)$ in Eq. (33) leads to a single, global optimum at $p_y = p_x$.

**Theorem 2** (Optimal Solution). *If the pair $(D^*, G^*)$ is a local Nash equilibrium for the Coulomb GAN objectives, then it is the global Nash equilibrium, and no other local Nash equilibria exist, and $G^*$ has output distribution $p_x = p_y$.*

*Proof.* See Appendix Section A.3. □

### 2.4.2 COULOMB GANS IN PRACTICE

To implement GANs in practice, we need learnable models for $G$ and $D$. We assume that our models for $G$ and $D$ are continuously differentiable with respect to their parameters and inputs. Toward this end, GANs are typically implemented as neural networks optimized by (some variant

of) gradient descent. Thus we may not find the optimal $G^*$ or $D^*$, since neural networks may suffer from capacity or optimization issues. Recent research indicates that the effect of local minima in deep learning vanishes with increasing depth (Dauphin et al., 2014; Choromanska et al., 2015; Kawaguchi, 2016), such that this limitation becomes less restrictive as our ability to train deep networks grows thanks to hardware and optimization improvements.

The main problem with learning Coulomb GANs is to approximate the potential function $\Phi$, which is a complex function in a high-dimensional space, since the potential can be very non-linear and non-smooth. When learning the discriminator, we must ensure that enough data is sampled and averaged over. We already lessened the non-linear function problem by using a low-dimensional Plummer kernel. But still, this kernel can introduce large non-linearities if samples are close to each other. It is crucial that the discriminator learns slow enough to accurately estimate the potential function which is induced by the current generator. The generator, in turn, must be even slower since it must be tracked by the discriminator. These approximation problems are supposed to be tackled by the research community in near future, which would enable optimal GAN learning.

The formulation of GAN learning as a potential field naturally solves the mode collapsing issue: the example described in Section A.1, where a normal GAN cannot get out of a local Nash equilibria is not a converged solution for the Coulomb GAN: If all probability mass of the generator lies in one of the modes, then both attracting forces from real-world samples located at the other mode as well as repelling forces from the over-represented generator mode will act upon the generator until it generates samples at the other mode as well.

## 3 EXPERIMENTS

In all of our experiments, we used a low-dimensional Plummer Kernel of dimensionality $d = 3$. This kernel both gave best computational performance and has low risk of running into numerical issues. We used a batch size of 128. To evaluate the quality of a GAN, the FID metric as proposed by Heusel et al. (2017) was calculated by using 50k samples drawn from the generator, while the training set statistics were calculated using the whole training set. We compare to BEGAN (Berthelot et al., 2017), DCGAN (Radford et al., 2016) and WGAN-GP (Gulrajani et al., 2017) both in their original version as well as when using the two-timescale update-rule (TTUR), using the settings from Heusel et al. (2017). We additionally compare to MMD-GAN (Li et al., 2017), which is conceptually very similar to the Coulomb GAN, but uses a Gaussian Kernel instead of the Plummer Kernel. We use the dataset-specific settings recommended in (Li et al., 2017) and report the highest FID score over the course of training. All images shown in this paper were produced with a random seed and not cherry picked. The implementation used for these experiments is available online[1]. The appendix Section A.5 contains an additional toy example demonstrating that Coulomb GANs do not suffer from mode collapse when fitting a simple Gaussian Mixture of 25 components.

### 3.1 IMAGE DATASETS

To demonstrate the ability of the Coulomb GAN to learn distributions in high dimensional spaces, we trained a Coulomb GAN on several popular image data sets: The cropped and centered images of celebrities from the Large-scale CelebFaces Attributes ("CelebA") data set (Liu et al., 2015), the *LSUN bedrooms* data set consists of over 3 million 64x64 pixel images of the bedrooms category of the large scale image database LSUN (Yu et al., 2015) as well as the CIFAR-10 data set. For these experiments, we used the DCGAN architecture (Radford et al., 2016) with a few modifications: our convolutional kernels all have a kernel size of 5x5, our random seed that serves as input to the generator has fewer dimensions: 32 for CelebA and LSUN bedrooms, and 16 for CIFAR-10. Furthermore, the discriminator uses twice as many feature channels in each layer as in the DCGAN architecture. For the Plummer kernel, $\epsilon$ was set to 1. We used the Adam optimizer with a learning rate of $10^{-4}$ for the generator and $5 \cdot 10^{-5}$ for the discriminator. To improve convergence performance, we used the $\tanh$ output activation function (LeCun et al., 1998). For regularization we used an L2 weight decay term with a weighting factor of $10^{-7}$. Learning was stopped by monitoring the FID metric (Heusel et al., 2017). Once learning plateaus, we scaled the learning rate down by a factor of 10 and let it continue once more until the FID plateaus. The results are reported in Table 1b, and generated

---

[1] www.github.com/bioinf-jku/coulomb_gan

images can be seen in Figure 2 and in the Appendix in Section A.7. Coulomb GANs tend to outperform standard GAN approaches like BEGAN and DCGAN, but are outperformed by the Improved Wasserstein GAN. However it is important to note that the Improved Wasserstein GAN used a more advanced network architecture based on ResNet blocks (Gulrajani et al., 2017), which we could not replicate due to runtime constraints. Overall, the low FID of Coulomb GANs stem from the fact that the images show a wide variety of different samples. E.g. on CelebA, Coulomb GAN exhibit a very wide variety of faces, backgrounds, eye colors and orientations. To further investigate how

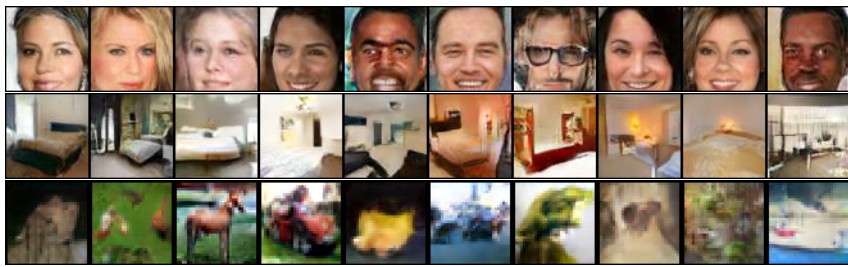

Figure 2: Images from a Coulomb GAN after training on CelebA (first row), LSUN bedrooms (second row) and CIFAR 10 (last row). Further examples are located in the appendix in Section A.7

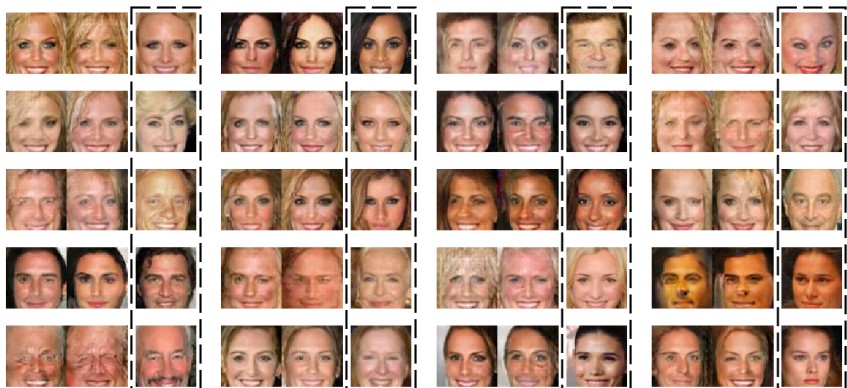

Figure 3: The most similar pairs found in batches of 1024 generated faces sampled from the Coulomb GAN, and the nearest neighbor from the training data shown as third image. Distances were calculated as Euclidean distances on pixel level.

much variation the samples generated by the Coulomb GAN contains, we followed the advice of Arora and Zhang (Arora & Zhang, 2017) to estimate the support size of the generator's distribution by checking how large a sample from the generator must be before we start generating duplicates. We were able to generate duplicates with a probability of around 50 % when using samples of size 1024, which indicates that the support size learned by the Coulomb GAN would be around 1M. This is a strong indication that the Coulomb GAN was able to spread out its samples over the whole target distribution. A depiction is included in Figure 3, which also shows the nearest neighbor in the training set of the generated images, confirming that the Coulomb GAN does not just memorize training images.

## 3.2 LANGUAGE MODELING

We repeated the experiments from Gulrajani et al. (2017), where Improved Wasserstein GANs (WGAN-GP) were trained to produce text samples after being trained on the Google Billion Word data set (Chelba et al., 2013), using the same network architecture as in the original publication. We use the Jensen-Shannon-divergence on 4-grams and 6-grams as an evaluation criterion. The results are summarized in Table 1a.

| data set | WGAN-GP | ours |
|---|---|---|
| 4 grams | 0.38 / 0.35 | 0.35 |
| 6 grams | 0.77 / 0.74 | 0.74 |

| data set | BEGAN | DCGAN | WGAN-GP | MMD | ours |
|---|---|---|---|---|---|
| CelebA | 29.2 / 28.5 | 21.4 / 12.5 | 4.8 / 4.2 | 63.2 | 9.3 |
| LSUN | 113 / 112 | 70.4 / 57.5 | 20.5 / 9.5 | 94.9 | 31.2 |
| CIFAR10 | - | - | 29.3 / 24.8 | 38.2 | 27.3 |

(a) Normalized Jensen-Shanon-Divergence for the Google Billion Word data. Values for WGAN-GP are without/with TTUR, taken from Heusel et al. (2017).

(b) Performance comparison in FID (lower is better) on different data sets. Values for all methods except Coulomb GAN and MMD-GAN are from without/with TTUR, taken from Heusel et al. (2017).

## 4    CONCLUSION

The Coulomb GAN is a generative adversarial network with strong theoretical guarantees. Our theoretical results show that the Coulomb GAN will be able to approximate the real distribution perfectly if the networks have sufficient capacity and training does not get stuck in local minima. Our results show that the potential field used by the Coulomb GAN far outperforms MMD based approaches due to its low-dimensional Plummer kernel, which is better suited for modeling probability density functions, and is very effective at eliminating the mode collapse problem in GANs. This is because our loss function forces the generated samples to occupy different regions of the learned distribution. In practice, we have found that Coulomb GANs are able to produce a wide range of different samples. However, in our experience, this sometimes leads to a small number of generated samples that are non-sensical interpolations of existing data modes. While these are sometimes also present in other GAN models (Radford et al., 2016), we found that our model produces such images at a slightly higher rate. This issue might be solved by finding better ways of learning the discriminator, as learning the correct potential field is crucial for the Coulomb GAN's performance. We also observed that increasing the capacity of the discriminator seems to always increase the generative performance. We thus hypothesize that the largest issue in learning Coulomb GANs is that the discriminator needs to approximate the potential field $\Phi$ very well in a high-dimensional space. In summary, instead of directly optimizing a criterion based on local differences of densities which can exhibit many local minima, Coulomb GANs are based on a potential field that has no local minima. The potential field is created by point charges in an analogy to electric field in physics. We have proved that if learning converges then it converges to the optimal solution if the samples can be moved freely. We showed that Coulomb GANs avoid mode collapsing, model the target distribution more truthfully than standard GANs, and do not overlook high probability regions of the target distribution.

ACKNOWLEDGMENTS

This work was supported by Zalando SE with Research Agreement 01/2016, Audi.JKU Deep Learning Center, Audi Electronic Venture GmbH, IWT research grant IWT150865 (Exaptation), H2020 project grant 671555 (ExCAPE) and NVIDIA Corporation. The authors would like to thank Philipp Renz for fruitful discussions.

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

## A  APPENDIX

### A.1  EXAMPLE OF CONVERGENCE TO MODE COLLAPSE IN CONVENTIONAL GANS

As an example of how a GAN can converge to a Nash Equilibrium that exhibits mode collapse, consider a target distribution that consists of two distinct/non-overlapping regions of support $C_1$ and $C_2$ that are distant from each other, i.e., the target probability is zero outside of $C_1$ and $C_2$. Further assume that 50 % of the probability mass is in $C_1$ and 50 % in $C_2$. Assume that the the generator has mode-collapsed onto $C_1$, which contains 100 % of the generator's probability mass. In this situation, the optimal discriminator classifies all points from $C_2$ as "real" (pertaining to the target distribution) by supplying an output of 1 for them (1 is the target for real samples and 0 the target for generated samples). Within $C_1$, the other region, the discriminator sees twice as many generated data points as real ones, as 100 % of the probability mass of the generator's distribution is in $C_1$, but only 50 % of the probability mass of the real data distribution. So one third of the points seen by the discriminator in $C_1$ are real, the other 2 thirds are generated. Thus, to minimize its prediction error for a proper objective (squared or cross entropy), the discriminator has to output $1/3$ for every point from $C_1$. The optimal output is even independent of the exact form of the real distribution in $C_1$. The generator will match the shape of the target distribution locally. If the shape is not matched, local gradients of the discriminator with respect to its input would be present and the generator would improve locally. If local improvements of the generator are no longer possible, the shape of the target distribution is matched and the discriminator output is locally constant. In this situation, the expected gradient of the discriminator is the zero vector, because it has reached an optimum. Since the discriminator output is constant in $C_1$ (and $C_2$), the generator's expected gradient is the zero vector, too. The situation is also stable even though we still have random fluctuations from the ongoing stochastic gradient (SGD) learning: whenever the generator produces data outside of (but close to) $C_1$, the discriminator can easily detect this and push the generator's samples back. Inside $C_1$, small deviations of the generator from the shape of the real distribution are detected by the discriminator as well, by deviating slightly from $1/3$. Subsequently, the generator is pushed back to the original shape. If the discriminator deviates from its optimum, it will also be forced back to its optimum. So overall, the GAN learning reached a local Nash equilibrium and has converged in the sense that the parameters fluctuate around the attractor point (fluctuations depend on learning rate, sample size, etc.). To achieve true mathematical convergence, Heusel et al. (2017) assume decaying learning rates to anneal the random fluctuations, similar to Robbins & Monro (1951) original convergence proof for SGD.

### A.2  PROOF OF THEOREM 1

We first recall Theorem 1:

**Theorem** (Convergence with low-dimensional Plummer kernel). *For $a, b \in \mathbb{R}^m$, $d \leqslant m - 2$, and $\epsilon > 0$ the densities $p_x(.)$ and $p_y(.)$ equalize over time when minimizing energy $F$ with the low-dimensional Plummer kernel by gradient descent. The convergence is faster for larger $d$.*

In a first step, we prove that for local maxima or local minima $a$ of $\rho$, the expression $\text{sign}(\nabla \cdot \mathbf{E}(a)) = \text{sign}(\rho(a))$ holds for $\epsilon$ small enough. For proving this equation, we apply the Laplace operator for spherical coordinates to the low-dimensional Plummer kernel. Using the result, we see that the integral $\nabla \cdot \mathbf{E}(a) = -\int \rho(b) \nabla_a^2 k(a, b) \, db$ is dominated by large negative values of $\nabla_a^2 k$ around $a$. These negative values can even be decreased by decreasing $\epsilon$. Therefore we can ensure by a small enough $\epsilon$ that at each local minimum and local maximum $a$ of $\rho$ $\text{sign}(\dot{\rho}(a)) = -\text{sign}(\rho(a))$. Thus, the maximal and minimal points of $\rho$ move toward zero.

In a second step, we show that new maxima or minima cannot appear and that the movement of $\Phi$ toward zero stops at zero and not earlier. Since $\rho$ is continuously differentiable, all points in environments of maxima and minima move toward zero. Therefore the largest $|\rho(a)|$ moves toward zero. We have to ensure that moving toward zero does not converge to a point apart from zero. We derive that the movement toward zero is lower bounded by $\dot{\rho}(a) = -\text{sign}(\rho(a))\lambda\rho^2(a)$. Thus, the movement slows down at $\rho(a) = 0$. Solving the differential equation and applying it to the maximum of the absolute value of $\rho$ gives $|\rho|_{\max}(t) = 1/(\lambda t + (|\rho|_{\max}(0))^{-1})$. Thus, $\rho$ converges to zero over time.

*Proof.* For $d = m - 2$, we have $\boldsymbol{\nabla}^2 k(\boldsymbol{a}, \boldsymbol{b}) = \delta(\boldsymbol{a} - \boldsymbol{b})$, where the theorem has already been proved for $\epsilon$ small enough (Hochreiter & Obermayer, 2005).

At each local minimum and local maximum $\boldsymbol{a}$ of $\rho$ we have $\boldsymbol{\nabla}_a \rho(\boldsymbol{a}) = 0$. Using the product rule for the divergence operator, Eq. (3) reduces to

$$\dot{\rho}(\boldsymbol{a}) = -\operatorname{sign}(\rho(\boldsymbol{a}))\, \rho(\boldsymbol{a})\, \boldsymbol{\nabla} \cdot \mathbf{E}(\boldsymbol{a}) \,. \tag{11}$$

The term $\boldsymbol{\nabla} \cdot \mathbf{E}(\boldsymbol{a})$ can be expressed as

$$\boldsymbol{\nabla} \cdot \mathbf{E}(\boldsymbol{a}) = -\boldsymbol{\nabla}_a^2 \Phi(\boldsymbol{a}) = -\int \rho(\boldsymbol{b})\, \boldsymbol{\nabla}_a^2 k(\boldsymbol{a}, \boldsymbol{b})\, \mathrm{d}\boldsymbol{b} \,. \tag{12}$$

We next consider $\boldsymbol{\nabla}_a^2 k(\boldsymbol{a}, \boldsymbol{b})$ for the low-dimensional Plummer kernel. We define the *spherical Laplace operator* in $(m-1)$ dimensions as $\boldsymbol{\nabla}_{S^{m-1}}^2$, then the Laplace operator in spherical coordinates is (Proposition 2.5 in Frye & Efthimiou (Efthimiou & Frye, 2014)):

$$\boldsymbol{\nabla}^2 = \frac{\partial^2}{\partial r^2} + \frac{m-1}{r}\frac{\partial}{\partial r} + \frac{m-1}{r^2}\boldsymbol{\nabla}_{S^{m-1}}^2 \,. \tag{13}$$

Note that $\boldsymbol{\nabla}_{S^{m-1}}^2$ only has second order derivatives with respect to the angles of the spherical coordinates.

With $r = \|\boldsymbol{a} - \boldsymbol{b}\|$ we obtain for the Laplace operator applied to the low-dimensional Plummer kernel:

$$\boldsymbol{\nabla}^2 k(\boldsymbol{a}, \boldsymbol{b}) = d\left(-\epsilon^2 m + (2 + d - m)\, r^2\right)(\epsilon^2 + r^2)^{-2-d/2} \,. \tag{14}$$

and in particular

$$\boldsymbol{\nabla}^2 k(\boldsymbol{a}, \boldsymbol{a}) = -m\, d\epsilon^{-(d+2)} \,. \tag{15}$$

For $d \leqslant m - 2$ we have $(2 + d - m) \leqslant 0$, and obtain

$$\boldsymbol{\nabla}^2 k(\boldsymbol{a}, \boldsymbol{b}) < 0 \,, \tag{16}$$

and

$$\frac{\partial}{\partial r}\boldsymbol{\nabla}^2 k(\boldsymbol{a}, \boldsymbol{b}) = d(2+d)\, r\left(\epsilon^2(2+m) + (-2-d+m)r^2\right)(\epsilon^2 + r^2)^{-3-d/2} > 0 \tag{17}$$

and

$$\frac{\partial}{\partial \epsilon}\boldsymbol{\nabla}^2 k(\boldsymbol{a}, \boldsymbol{b}) = d(2+d)\, \epsilon\left(\epsilon^2 m + (-4-d+m)r^2\right)(\epsilon^2 + r^2)^{-3-d/2} > 0 \,. \tag{18}$$

Therefore, $\boldsymbol{\nabla}^2 k(\boldsymbol{a}, \boldsymbol{b})$ is negative with minimum $-md\epsilon^{-(d+2)}$ at $r = 0$ and increasing with $r$ and increasing with $\epsilon$ for $d \leqslant m - 4$. For $d = m - 3$ we have to restrict in the following the sphere $S_\tau(\boldsymbol{a})$ to $\tau < \sqrt{m}\epsilon$ and ensure increase of $\boldsymbol{\nabla}^2 k(\boldsymbol{a}, \boldsymbol{b})$ with $\epsilon$.

If $\rho(\boldsymbol{b}) \neq 0$, then we define a sphere $S_\tau(\boldsymbol{a})$ with radius $\tau$ around $\boldsymbol{a}$ for which holds $\operatorname{sign}(\rho(\boldsymbol{b})) = \operatorname{sign}(\rho(\boldsymbol{a}))$ for each $\boldsymbol{b} \in S_\tau(\boldsymbol{a})$. Note that $\boldsymbol{\nabla}^2 k(\boldsymbol{a}, \boldsymbol{b})$ is continuous differentiable. We have

$$\boldsymbol{\nabla} \cdot \mathbf{E}(\boldsymbol{a}) = -\int \rho(\boldsymbol{b})\, \boldsymbol{\nabla}_a^2 k(\boldsymbol{a}, \boldsymbol{b})\, \mathrm{d}\boldsymbol{b} = \tag{19}$$

$$-\int_{S_\tau(\boldsymbol{a})} \rho(\boldsymbol{b})\, \boldsymbol{\nabla}_a^2 k(\boldsymbol{a}, \boldsymbol{b})\, \mathrm{d}\boldsymbol{b} - \int_{T \setminus S_\tau(\boldsymbol{a})} \rho(\boldsymbol{b})\, \boldsymbol{\nabla}_a^2 k(\boldsymbol{a}, \boldsymbol{b})\, \mathrm{d}\boldsymbol{b} \,.$$

We bound $\boldsymbol{\nabla}^2 k(\boldsymbol{a}, \boldsymbol{b})$ by

$$0 > \boldsymbol{\nabla}^2 k(\boldsymbol{a}, \boldsymbol{b}) = d\left(-\epsilon^2 m + (2 + d - m)\, r^2\right)(\epsilon^2 + r^2)^{-2-d/2} > d(2 + d - m)\, r^{-2-d} \,. \tag{20}$$

Using $\tau$, we now bound $\left|\int_{T \setminus S_\tau(\boldsymbol{a})} \rho(\boldsymbol{b})\, \boldsymbol{\nabla}_a^2 k(\boldsymbol{a}, \boldsymbol{b})\, \mathrm{d}\boldsymbol{b}\right|$ independently from $\epsilon$, since $\rho$ is a difference of distributions. For small enough $\epsilon$ we can ensure

$$\left|\int_{S_\tau(\boldsymbol{a})} \rho(\boldsymbol{b})\, \boldsymbol{\nabla}_a^2 k(\boldsymbol{a}, \boldsymbol{b})\, \mathrm{d}\boldsymbol{b}\right| > \left|\int_{T \setminus S_\tau(\boldsymbol{a})} \rho(\boldsymbol{b})\, \boldsymbol{\nabla}_a^2 k(\boldsymbol{a}, \boldsymbol{b})\, \mathrm{d}\boldsymbol{b}\right| \,. \tag{21}$$

Therefore we have

$$\text{sign}(\boldsymbol{\nabla} \cdot \mathbf{E}(\boldsymbol{a})) = \text{sign}(\rho(\boldsymbol{a})) . \tag{22}$$

Therefore we have at each local minimum and local maximum $\boldsymbol{a}$ of $\rho$

$$\text{sign}(\dot{\rho}(\boldsymbol{a})) = -\text{sign}(\rho(\boldsymbol{a})) . \tag{23}$$

Therefore the maximal and minimal points of $\rho$ move toward zero. Since $\rho$ is continuously differentiable as is the field, also the points in an environment of the maximal and minimal points move toward zero. Points that are not in an environment of the maximal or minimal points cannot become maximal points in an infinitesimal time step.

Since the contribution of $\boldsymbol{a}$ environment $S_\tau(\boldsymbol{a})$ dominates the integral Eq. (19), for $\epsilon$ small enough there exists a positive $0 < \lambda$ globally for all minima and maxima as well as for all time steps for which holds:

$$|\boldsymbol{\nabla} \cdot \mathbf{E}(\boldsymbol{a})| > \lambda \, |\rho(\boldsymbol{a})| . \tag{24}$$

The factor $\lambda$ depends on $k$ and on the initial $\rho$. $\lambda$ is proportional to $d$. Larger $d$ lead to larger $|\boldsymbol{\nabla} \cdot \mathbf{E}(\boldsymbol{a})|$ since the maximum or minimum $\rho(\boldsymbol{a})$ is upweighted. There might exist initial conditions $\rho$ for which $\lambda \to 0$, e.g. for infinite many maxima and minima, but they are impossible in our applications.

Therefore maximal or minimal points approach zero faster or equal than given by

$$\dot{\rho}(\boldsymbol{a}) = -\text{sign}(\rho(\boldsymbol{a})) \, \lambda \, \rho^2(\boldsymbol{a}) . \tag{25}$$

In particular this differential equation dominates the global maximum $|\rho|_{\max}$ of $|\rho(.)|$. Solving the differential equation gives that at least

$$|\rho|_{\max}(t) = \frac{1}{\lambda \, t + (|\rho|_{\max}(0))^{-1}} . \tag{26}$$

Thus $d$ influences the worst case rate of convergence, where larger $d$ with $d \leqslant m - 2$ leads to faster worst case convergence.

Consequently, $\rho$ converges to the zero function over time, that is, $p_x(.)$ becomes equal to $p_y(.)$. $\quad\square$

## A.3 PROOF OF THEOREM 2

We first recall Theorem 2:

**Theorem** (Optimal Solution). *If the pair $(D^*, G^*)$ is a local Nash equilibrium for the Coulomb GAN objectives, then it is the global Nash equilibrium, and no other local Nash equilibria exist, and $G^*$ has output distribution $p_x = p_y$.*

*Proof.* $(D^*, G^*)$ being in a local Nash equilibrium means that $(D^*, G^*)$ fulfills the two conditions

$$D^* = \underset{D \in U(D^*)}{\arg\min} \, \mathcal{L}_D(D; G^*) \quad \text{and} \quad G^* = \underset{G \in U(G^*)}{\arg\min} \, \mathcal{L}_G(G; D^*) \tag{27}$$

for some neighborhoods $U(D^*)$ and $U(G^*)$. For Coulomb GANs that means, $D^*$ has learned the potential $\Phi$ induced by $G^*$ perfectly, because $\mathcal{L}_D$ is convex in $D$, thus if $D^*$ is optimal within an neighborhood $U(D^*)$, it must be the global optimum. This means that $G^*$ is directly minimizing $\mathcal{L}_G(G; D) = -\frac{1}{2}\mathrm{E}_{p_z}(\Phi(G(z)))$. The Coulomb potential energy is according to Eq. (7)

$$F(\rho) = \frac{1}{2} \int \rho(\boldsymbol{a}) \Phi(\boldsymbol{a}) \mathrm{d}\boldsymbol{a} = \frac{1}{2} \int p_y(\boldsymbol{a}) \Phi(\boldsymbol{a}) \mathrm{d}\boldsymbol{a} - \frac{1}{2} \int p_x(\boldsymbol{a}) \Phi(\boldsymbol{a}) \mathrm{d}\boldsymbol{a} . \tag{28}$$

Only the samples from $p_x$ stem from the generator, where $p_x(\boldsymbol{a}) = \int \delta(\boldsymbol{a} - G(z)) p_z(z) \mathrm{d}z$. Here $\delta$ is the $\delta$-distribution centered at zero. The part of the energy which depends on the generator is

$$-\frac{1}{2} \int p_x(\boldsymbol{a}) \, \Phi(\boldsymbol{a}) \, \mathrm{d}\boldsymbol{a} = -\frac{1}{2} \int \int \delta(\boldsymbol{a} - G(z)) \, p_z(z) \, \mathrm{d}z \, \Phi(\boldsymbol{a}) \, \mathrm{d}\boldsymbol{a} \tag{29}$$

$$= -\frac{1}{2} \int \left( \int \delta(\boldsymbol{a} - G(z)) \, \Phi(\boldsymbol{a}) \, \mathrm{d}\boldsymbol{a} \right) p_z(z) \, \mathrm{d}z$$

$$= -\frac{1}{2} \int p_z(z) \, \Phi(G(z)) \, \mathrm{d}z = -\frac{1}{2} \mathrm{E}_{p_z}(\Phi(G(z))) .$$

Theorem 1 guarantees that there are no other local minima except the global one when minimizing $F$. $F$ has one minimum, $F = 0$, which implies $\Phi(\boldsymbol{a}) = 0$ and $\rho(\boldsymbol{a}) = 0$ for all $\boldsymbol{a}$, therefore also $p_y = p_x$ according to Theorem 1. Each $\Phi(\boldsymbol{a}) \neq 0$ would mean there exist potential differences which in turn would cause forces on generator samples that allow to further minimize the energy. Since we assumed that the generator can reach the minimum $p_y = p_x$ for any $p_y$, it will be reached by local (stepwise) optimization of $-\frac{1}{2}\mathrm{E}_{p_z}\left(\Phi(G(\boldsymbol{z}))\right)$ with respect to $G$. Since the pair $(D^*, G^*)$ is optimal within their neighborhood, the generator has reached this minimum as there is not other local minimum than the global one. Therefore $G^*$ has model density $p_x$ with $p_y = p_x$. The convergence point is a global Nash equilibrium, because there is no approximation error and zero energy $F = 0$ is a global minimum for discriminator and generator, respectively. Theorem 1 ensures that other local Nash equilibria are not possible. □

### A.4 COULOMB EQUATIONS IN THE CASE OF FINITE SAMPLES

GANs are sample-based, that is, samples are drawn from the model for learning (Hochreiter & Obermayer, 2005; Gutmann & Hyvärinen, 2012). Typically this is done in mini-batches, where each mini-batch consists of two sets of samples, the target samples $\mathcal{Y} = \{\boldsymbol{y}_i | i = 1 \dots N_y\}$, and the model samples $\mathcal{X} = \{\boldsymbol{x}_i | i = 1 \dots N_x\}$.

For such finite samples, i.e. point charges, we have to use delta distributions to obtain unbiased estimates of the the model distribution $p_x(.)$ and the target distribution $p_y(.)$:

$$\hat{p}_y(\boldsymbol{a}; \mathcal{Y}) = \frac{1}{N_y} \sum_{i=1}^{N_y} \delta(\boldsymbol{a} - \boldsymbol{y}_i) \ , \quad \hat{p}_x(\boldsymbol{a}; \mathcal{X}) = \frac{1}{N_x} \sum_{i=1}^{N_x} \delta(\boldsymbol{a} - \boldsymbol{x}_i) \ , \quad \hat{\rho}(\boldsymbol{a}; \mathcal{X}, \mathcal{Y}) = p_y(\boldsymbol{a}; \mathcal{Y}) - p_x(\boldsymbol{a}; \mathcal{X}) \ , \tag{30}$$

where $\delta$ is the Dirac $\delta$-distribution centered at zero. These are unbiased estimates of the underlying distribution, as can be seen by:

$$\mathrm{E}_{\mathcal{X}}\left(\frac{1}{N_x} \sum_{i=1}^{N_x} \delta(\boldsymbol{a} - \boldsymbol{x}_i)\right) = \frac{1}{N_x} \sum_{i=1}^{N_x} \mathrm{E}_{\boldsymbol{x}_i}\left(\delta(\boldsymbol{a} - \boldsymbol{x}_i)\right) = \frac{1}{N_x} \sum_{i=1}^{N_x} p_x(\boldsymbol{a}) = p_x(\boldsymbol{a}) \ . \tag{31}$$

In the rest of the paper, we will drop the explicit parameterization with $\mathcal{X}$ and $\mathcal{Y}$ for all estimates to unclutter notation, and instead just use the hat sign to denote estimates. In the same fashion as for the distributions, when we use fixed samples $\mathcal{X}$ and $\mathcal{Y}$, we obtain the following unbiased estimates for the potential, energy and field given by Eq. (5), Eq. (6), and Eq. (7):

$$\hat{\Phi}(\boldsymbol{a}) = \frac{1}{N_y} \sum_{i=1}^{N_y} k(\boldsymbol{a}, \boldsymbol{y}_i) - \frac{1}{N_x} \sum_{i=1}^{N_x} k(\boldsymbol{a}, \boldsymbol{x}_i) \ , \tag{32}$$

$$\hat{F}(\rho) = \frac{1}{2}\left(\frac{1}{N_y^2} \sum_{i=1}^{N_y}\sum_{j=1}^{N_y} k(\boldsymbol{y}_i, \boldsymbol{y}_j) - \frac{2}{N_y N_x} \sum_{i=1}^{N_y}\sum_{j=1}^{N_x} k(\boldsymbol{y}_i, \boldsymbol{x}_j) + \frac{1}{N_x^2} \sum_{i=1}^{N_x}\sum_{j=1}^{N_x} k(\boldsymbol{x}_i, \boldsymbol{x}_j)\right) \tag{33}$$

$$= \frac{1}{2}\left(\frac{1}{N_y} \sum_{i=1}^{N_y} \hat{\Phi}(\boldsymbol{y}_i) - \frac{1}{N_x} \sum_{i=1}^{N_x} \hat{\Phi}(\boldsymbol{x}_i)\right) \ ,$$

$$\hat{\mathbf{E}}(\boldsymbol{a}) = -\boldsymbol{\nabla}_a \hat{\Phi}(\boldsymbol{a}) = -\frac{1}{N_y} \sum_{i=1}^{N_y} \boldsymbol{\nabla}_a k(\boldsymbol{a}, \boldsymbol{y}_i) + \frac{1}{N_x} \sum_{i=1}^{N_x} \boldsymbol{\nabla}_a k(\boldsymbol{a}, \boldsymbol{x}_i) \tag{34}$$

$$\hat{\mathbf{E}}(\boldsymbol{y}_i) = -N_y \boldsymbol{\nabla}_{y_i} \hat{F}(\rho) \ , \quad \hat{\mathbf{E}}(\boldsymbol{x}_i) = N_x \boldsymbol{\nabla}_{x_i} \hat{F}(\rho) \ .$$

These are again unbiased, e.g.:

$$
\begin{aligned}
\mathrm{E}_{\mathcal{X},\mathcal{Y}}(\hat{\Phi}(\boldsymbol{a})) &= \frac{1}{N_y} \sum_{i=1}^{N_y} E_{\boldsymbol{y}_i}\left(k(\boldsymbol{a}, \boldsymbol{y}_i)\right) - \frac{1}{N_x} \sum_{i=1}^{N_x} E_{\boldsymbol{x}_i}\left(k(\boldsymbol{a}, \boldsymbol{x}_i)\right) \qquad (35) \\
&= \frac{1}{N_y} \sum_{i=1}^{N_y} \int p_y(\boldsymbol{y}_i)\, k(\boldsymbol{a}, \boldsymbol{y}_i)\, \mathrm{d}\boldsymbol{y}_i - \frac{1}{N_x} \sum_{i=1}^{N_x} \int p_x(\boldsymbol{x}_i)\, k(\boldsymbol{a}, \boldsymbol{x}_i)\, \mathrm{d}\boldsymbol{x}_i \\
&= \int p_y(\boldsymbol{y})\, k(\boldsymbol{a}, \boldsymbol{y})\, \mathrm{d}\boldsymbol{y} - \int p_x(\boldsymbol{x})\, k(\boldsymbol{a}, \boldsymbol{x})\, \mathrm{d}\boldsymbol{x} \\
&= \int (p_y(\boldsymbol{x}) - p_x(\boldsymbol{x}))\, k(\boldsymbol{a}, \boldsymbol{x})\, \mathrm{d}\boldsymbol{x} = \int \rho(\boldsymbol{x})\, k(\boldsymbol{a}, \boldsymbol{x})\, \mathrm{d}\boldsymbol{x} = \Phi(\boldsymbol{a})
\end{aligned}
$$

If we draw samples of infinite size, all these expressions for a fixed sample size lead to the equivalent statements for densities. The sample-based formulation, that is, point charges in physical terms, can only have local energy minima or maxima at locations of samples (Dembo & Zeitouni, 1988). Furthermore the field lines originate and end at samples, therefore the field guides model samples $\boldsymbol{x}$ toward real world samples $\boldsymbol{y}$, as depicted in Figure 1. The factors $N_y$ and $N_x$ in the last equations arise from the fact that $-\boldsymbol{\nabla}_a F$ gives the force which is applied to a sample with charge. A sample $\boldsymbol{y}_i$ is positively charged with $1/N_y$ and follows $-\boldsymbol{\nabla}_{y_i} F$ while a sample $\boldsymbol{x}_i$ is negatively charged with $-1/N_x$ and therefore follows $-\boldsymbol{\nabla}_{x_i} F$, too. Thus, following the force induced on a sample by the field is equivalent to gradient descent of the energy $F$ with respect to samples $\boldsymbol{y}_i$ and $\boldsymbol{x}_i$.

## A.5 MIXTURE OF GAUSSIANS

We use the synthetic data set introduced by Lim & Ye (2017) to show that Coulomb GANs avoid mode collapse and that all modes of the target distribution are captured by the generative model. This data set comprises 100K data points drawn from a Gaussian mixture model of 25 components which are spread out evenly in the range $[-21, 21] \times [-21, 21]$, with each component having a variance of 1. To make results comparable with Lim & Ye (2017), the Coulomb GAN used a discriminator network with 2 hidden layers of 128 units, however we avoided batch normalization by using the ELU activation function (Clevert et al., 2016). We used the Plummer kernel in 3 dimensions ($d = 3$) with an epsilon of 3 ($\epsilon = 3$) and a learning rate of 0.01, both of which were exponentially decayed during the 1M update steps of the Adam optimizer.

As can be seen in Figure 4, samples from the learned Coulomb GAN very well approximate the target distribution. All components of the original distribution are present at the model distribution at approximately the correct ratio, as shown in Figure 5. Moreover, the generated samples are distributed approximately according to the same spread for each component of the real world distribution. Coulomb GANs outperform other compared methods, which either fail to learn the distribution completely, ignore some of the modes, or do not capture the within-mode spread of a Gaussian. The Coulomb GAN is the only GAN approach that manages to avoid a within-cluster collapse leading to insufficient variance within a cluster.

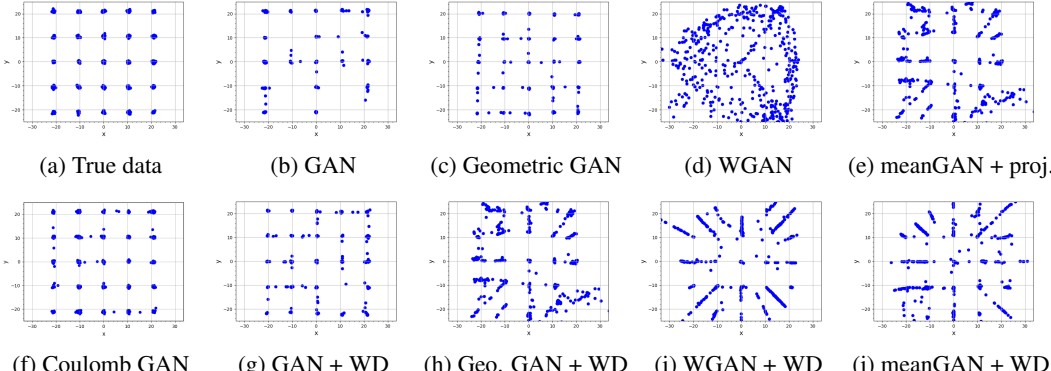

Figure 4: Scatter plots of generated samples from different GAN variants for the mixture of 25 Gaussians and the true data distribution. "WD" indicates weight decay and "proj." means projection. Results and images for all methods except the Coulomb GAN are taken from Lim & Ye (2017).

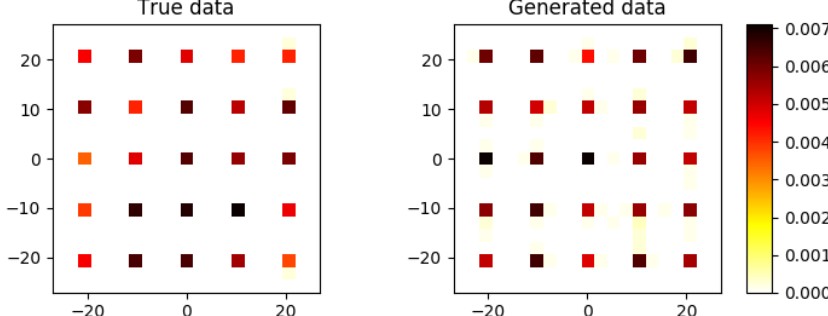

Figure 5: 2D histogram of the density of generated and the training st data for the mixture of 25 Gaussians. For constructing the histogram, 10k samples were drawn from the target and the model distribution. The Coulomb GAN captures the underlying distribution well, does not miss any modes, and places almost all probability mass on the modes. Only the Coulomb GAN captured the within-mode spread of the Gaussians.

## A.6 PSEUDOCODE FOR COULOMB GANS

The following gives the pseudo code for training GANs. Note that when calculating the derivative of $\hat{\Phi}(\boldsymbol{a}_i; \mathcal{X}, \mathcal{Y})$, it is important to only derive with respect to $\boldsymbol{a}$, and not wrt. $\mathcal{X}, \mathcal{Y}$, even if it can happen that e.g. $\boldsymbol{a} \in \mathcal{X}$. In frameworks that offer automatic differentiation such as Tensorflow or Theano, this means stopping the possible gradient back-propagation through those parameters.

---

**Algorithm 1** Minibatch stochastic gradient descent training of Coulomb GANs for updating the the discriminator weights $\boldsymbol{w}$ and the generator weights $\boldsymbol{\theta}$.

---

**while** Stopping criterion not met **do**
- Sample minibatch of $N_x$ training samples $\{\boldsymbol{x}_1, \ldots, \boldsymbol{x}_{N_x}\}$ from training set
- Sample minibatch of $N_y$ generator samples $\{\boldsymbol{y}_1, \ldots, \boldsymbol{y}_{N_y}\}$ from the generator
- Calculate the gradient for the discriminator weights:

$$d\boldsymbol{w} \leftarrow \nabla_{\boldsymbol{w}} \left[ \frac{1}{2} \sum_{i=1}^{N_x} \left( D(\boldsymbol{x}_i) - \hat{\Phi}(\boldsymbol{x}_i) \right)^2 + \frac{1}{2} \sum_{i=1}^{N_y} \left( D(\boldsymbol{y}_i) - \hat{\Phi}(\boldsymbol{y}_i) \right)^2 \right]$$

- Calculate the gradient for the generator weights:

$$d\boldsymbol{\theta} \leftarrow \nabla_{\boldsymbol{\theta}} \left[ -\frac{1}{2} \frac{1}{N_x} \sum_{i=1}^{N_x} D\left(\boldsymbol{x}_i\right) \right]$$

- Update weights according to optimizer rule (e.g. Adam):

$$\boldsymbol{w}_{n+1} = \boldsymbol{w}_n + \texttt{ADAM}(d\boldsymbol{w}, n)$$
$$\boldsymbol{\theta}_{n+1} = \boldsymbol{\theta}_n + \texttt{ADAM}(d\boldsymbol{\theta}, n)$$

**end while**

---

### A.7 More Samples from Coulomb GANs

CelebA

Images from a Coulomb GAN after training on CelebA data set. The low FID stems from the fact that the images show a wide variety of different faces, backgrounds, eye colors and orientations.

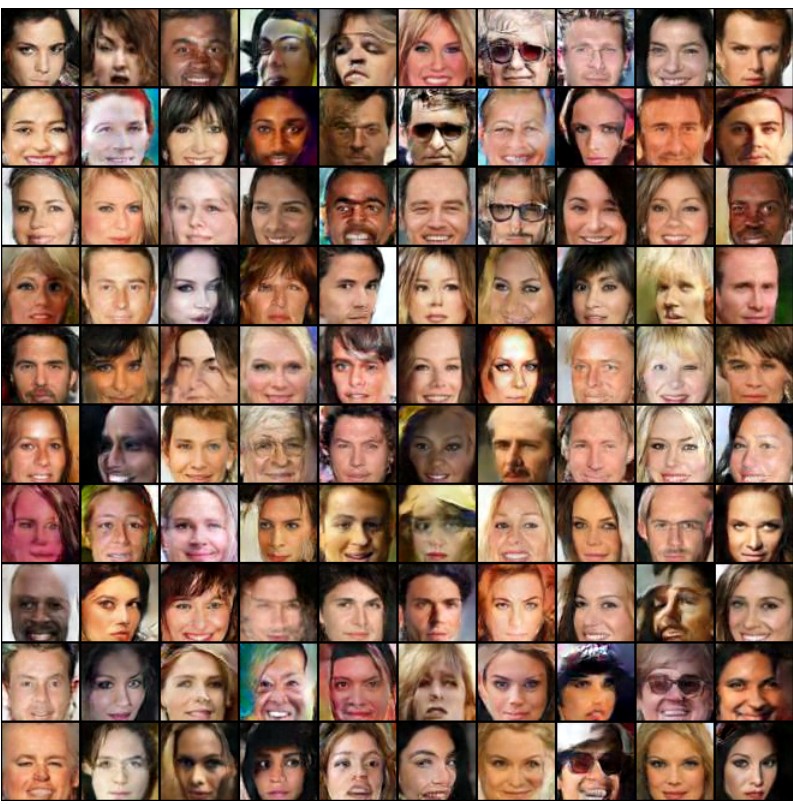

LSUN BEDROOMS

Images from a Coulomb GAN after training on the LSUN bedroom data set.

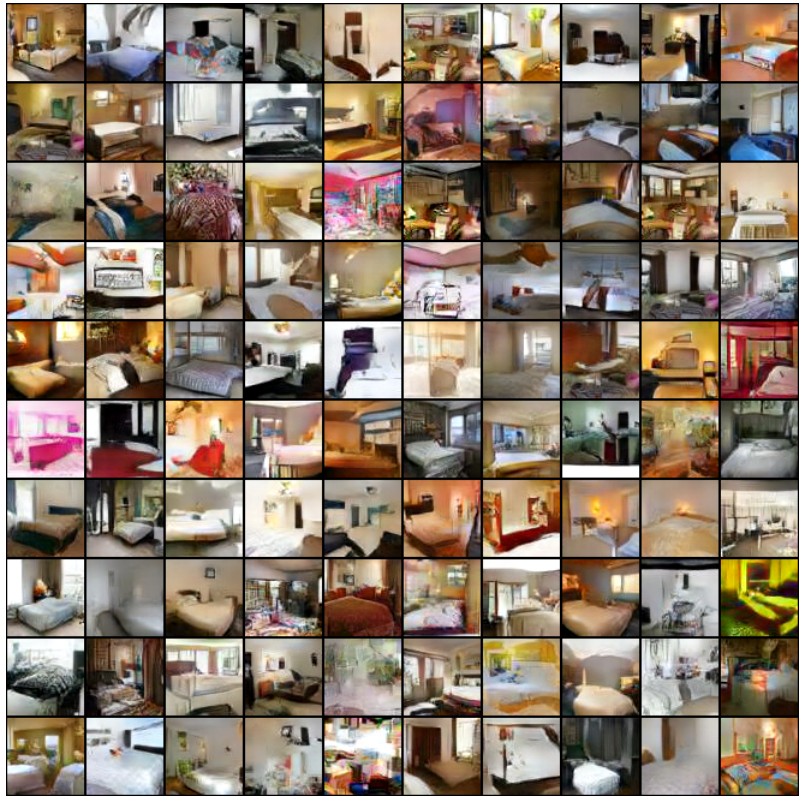

CIFAR 10

Images from a Coulomb GAN after training on the CIFAR 10 data set.

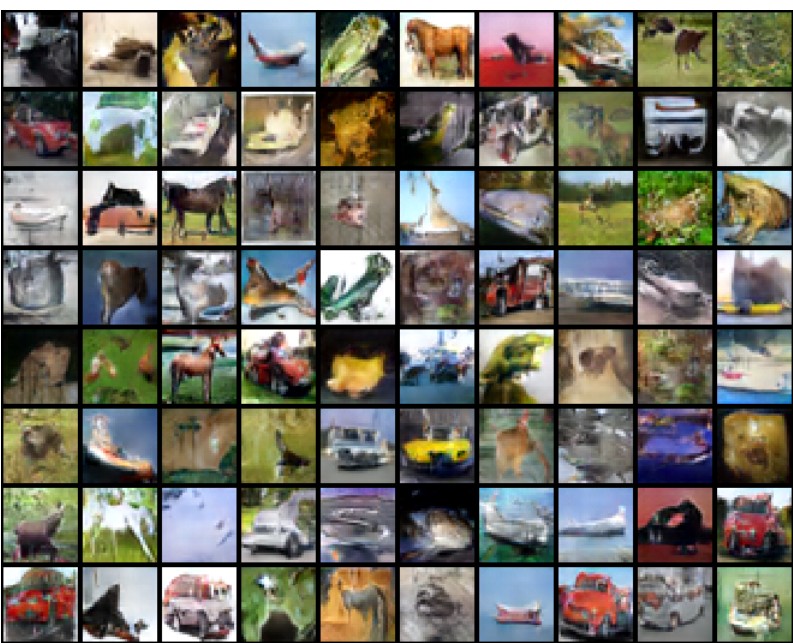

