# OpenReview forum: "Coulomb GANs: Provably Optimal Nash Equilibria via Potential Fields"
_ICLR.cc/2018/Conference — Accept (Poster)_

### Official Review · AnonReviewer2 · 2017-11-27
**Several points should be clarified**

**Rating:** 5
**Confidence:** 4

**Review:**


In this paper, the authors interpret the training of GAN by potential field and inspired from which to provide new training procedure for GAN. They claim that under the condition that global optima are achieved for discriminator and generator in each iteration, the Coulomb GAN converges to the global solution.

I think there are several points need to be addressed.

1, I agree that the "model collapsing" is due to converging to a local Nash Equilibrium. However, there are more reasons besides the drawback of the loss function, which is emphasized in the paper. Leave the stochastic gradient descent optimization algorithm apart (since most of the neural networks are trained in this way), the parametrization and the richness of discriminator family play a vital role in the model collapsing issue. In fact, even with KL-divergence in which log operation is involved, if one can select reasonable parametrization, e.g., directly handling in function space, the saddle point optimization is convex-concave, which means under the same assumption made in the paper, there is only one global Nash Equilibrium. On the other hand, the richness of the discriminator also important in the training of GAN. I did not get the point about the drawback of III. If indeed as the paper considered in the ideal case, the discriminator is rich enough, III cannot happen.

The model collapsing is not just because loss function in training GAN. It is caused by the twist of these three issues listed above. Modifying the loss can avoid partially model collapsing, however, it is not appropriate to claim that the proposed algorithm is 'provable'. The assumption in this paper is too restricted, and the discussion is unfair to the existing variants of GAN, e.g., GMMN or Wasserstein GAN, which under some assumptions, there is also only one global Nash Equilibrium.

2, In the training procedure, the discriminator family is important as we discussed. The paper claims that the reason to introduce the extra discriminator is reducing variance. However, such parametrization will introduce bias too. The bias and variance tradeoff should be explicitly discussed here. Ideally, it should contain all the functions formed with Plummer kernel, but not too large (otherwise, it will increase the sample complexity.). Which function family used in the paper is not clear.


3, As the authors already realized, the GMMN is one closely related model. It will be more convincing to add the comparison with GMMN.

In sum, this paper provides an interesting perspective modeling GAN from the potential field, however, there are several issues need to be addressed. I expect to see the reply of the authors regarding the mentioned issues.

---

> ### Author Response · Authors · 2018-01-02
> **Response to review**
>
> We'd like to thank the reviewer for this in-depth and constructive review, it was a great help in improving our manuscript. The major changes in the new version of the text are:
>
> * Clarified notion of Nash Equilibria: reformulated Theorem 2 in function space (your point 1)
> * Clarified bias/variance issues (your point 2)
> * Added comparison to MMD GAN (your point 3)
>
> In hindsight, we didn't outline the contribution that Coulomb GANs make as clear as we could have. As a result of this review, we have rewritten large portions of Section 2 and strongly clarified some of our main statements. We were able to  reformulate Theorem 2 in a much more precise way. We think that the new version of the text is much improved, and we hope it clears up the items you mentioned.
>
> To your specific points:
>
> 1. Thank you, this comment was very helpful, and lead us to reformulate some of our claims in a clearer way. We agree that loss functions are not the only culprit for unsuccessful GAN learning, and that all practical learning approaches - where generator/discriminator are parametrized models and learned by SGD - introduce a whole lot of convergence and local optimality problems in GANs. But more fundamentally the choice of the loss function might introduce bad local Nash equlibria already in the "theoretical" function space. This fundamental issue is - to the best of our knowledge - not explored in the current literature, neither in the context of Wasserstein GANs nor in GMMN losses and we are not sure if the absence of local Nash equilibria in function space could be proven for those cases. This issue has fundamental implications for all GAN architectures.  Therefore our work aims to be more than "just another cool GAN“, but hopefully furthers the theoretical understanding of GANs in general.
> We think that the main contribution of Coulomb GANs is to provide a loss function for GAN learning with the mathematically rigorous guarantee that no such local Nash Equlibria *IN FUNCTION SPACE* exist.  We think this is a crucial issue that has not received proper attention yet in order to put scientific GAN research on a solid rigorous ground. We are not aware of any other paper that provides such a strong claim as our Theorem 2. Neither WGAN nor MMD-based approaches have made this claim and we are not sure that a corresponding claim for them would be provable at all.
> We hope you will appreciate our newly written section 2.1, where we discuss in more depth and mathematical precision what we mean by "local nash equlibrium in function space", and how it differs from looking at things in probability-distribution space or parameter space.
> With that said, we fully agree that for all practical purposes the choice of rich discriminators (and the parametrization in general) is highly important for good empirical performance. However, that topic is not the main point we are  trying to investigate.
>
> 2. You are right, thank you for this head's up! There are two kinds of approximation here: First, we approximate the potential Phi using a mini-batch specific \hat{Phi}. The newer version of the paper discusses the properties of this approximation. Concretely, we show in the appendix that the estimate is unbiased, and explicitly mention the drawback of its high variance in the main text (Section 2.4).
> Secondly, as you correctly stated, we learn Phi with a neural network (the discriminator) to reduce the high variance of the mini-batch \hat{Phi}. With this, we run into the usual the bias/variance tradeoff of Machine Learning: trading overfitting against underfitting. And we absolutely agree that finding a good discriminator (that is able to learn the potential field correctly) is vital. Thankfully, in GAN learning we can always sample new generator data in each mini-batch, so overfitting on those is not too much of an issue, but we could still overfit on the real-world training data. This could lead to local Nash equilibria in parameter space. Therefore, we tried to be more explicit in the new version of the text that our analysis focuses on the space of functions, and we explicitly mention that neural network learning is vulnerable to issues such as over/underfitting (again in Section 2.4).
>
> 3. Thank you for the suggestion, we have added this comparison: The original GMMN approach is computationally very expensive to run on the typical GAN datasets, and was recently improved upon by the MMD-GAN model [Li et al, NIPS 2017]. Most importantly, MMD GAN extends the GMMN approach to a learnable discriminator, which makes the approach better and feasible for larger datasets (& very similar to Coulomb GAN's discriminator, presumably with the same advantage of reducing variance). In their paper, Li et al. show that MMD GAN outperforms GMMN on all tested datasets. We thus added a comparison to MMD-GAN to the current revision of the manuscript.

---

### Official Review · AnonReviewer1 · 2017-11-30
**Good step forward in improving GANs**

**Rating:** 7
**Confidence:** 3

**Review:**

The authors draw from electrical field dynamics and propose to formulate the GAN learning problem in a way such that generated samples are attracted to training set samples, but repel each other. Optimizing this formulation using gradient descent can be proven to yield only one optimal global Nash equilibrium, which the authors claim allows Coulomb GANs to overcome the "mode collapse" issue. Experimental results are reported on image and language modeling tasks, and show that the model can produce very diverse samples, although some samples can consist of somewhat nonsensical interpolations.

This is a good, well-written paper. It is technically rigorous and empirically convincing. Overall, it presents an interesting approach to overcome the mode collapse problem with GANs.

The image samples presented -- although of high variability -- are not of very high quality, though, and I somewhat disagree with the claim that "Coulomb GAN was able to efficiently learn the whole distribution" (Sec 3.1). At best, it seems to me that the new objective does in fact force the generator to concentrate efforts on learning over the full support of the data distribution, but the lower quality samples and sometimes somewhat bad interpolations seem to suggest to me that it is *not* yet doing so very "efficiently".

Nonetheless, I think this is an important step forward in improving GANs, and should be accepted for publication.

Note: I did not check all the proofs in the appendix.

---

> ### Author Response · Authors · 2018-01-02
> **Response to review**
>
> Thank you for your review, and thanks for the „important step forward in improving GANs“. We appreciate your positive feedback. We agree with your assessment that our objective forces the generator to concentrate efforts on learning over the full support at the cost of somewhat bad interpolations, and toned down the statement about learning "efficiently" in the new update of the paper.
> To see how Coulomb GANs perform when contrasted with similar approaches that aim to learn the full support of the distribution, we added a new comparison with MMD approaches. It turns out that Coulomb GANs are more efficient than MMD GANs.

---

### Official Review · AnonReviewer3 · 2017-12-01
**The paper introduces a new way to address GAN training problems, equating it to potential fields and taking inspiration from electrostatics.**

**Rating:** 7
**Confidence:** 2

**Review:**

The paper takes an interesting approach to solve the existing problems of GAN training, using Coulomb potential for addressing the learning problem. It is also well written with a clear presentation of the motivation of the problems it is trying to address, the background and proves the optimality of the suggested solution. My understanding and validity of the proof is still an educated guess. I have been through section A.2 , but I'm unfamiliar with the earlier literature on the similar topics so I would not be able to comment on it.

Overall, I think this is a good paper that provides a novel way of looking at and solving problems in GANs. I just had a couple of points in the paper that I would like some clarification on :

* In section 2.2.1 : The notion of the generated a_i not disappearing is something I did not follow. What does it mean for a generated sample to "not disappear" ? and this directly extends to the continuity equation in (2).

* In section 1 : in the explanation of the 3rd problem that GANs exhibit i.e.  the generator not being able to generalize the distribution of the input samples, I was hoping if you could give a bit more motivation as to why this happens. I don't think this needs to be included in the paper, but would like to have it for a personal clarification.

---

> ### Author Response · Authors · 2018-01-02
> **Response to Review**
>
> We thank the reviewer for their encouraging review, appreciate the positive feedback. For your questions:
>
> * Thanks for pointing out that our explanation was not clear enough. An a_i is associated with a particular random variable z_i of the generator which is mapped by the generator to a_i. If the generator changes, then the same random variable z_i is mapped to another a_i'. That is a_i moved to a_i'. We have explained this more clearly in the current revision.
>
> *  We have tried to explain this better in the new version of the text (even if you said it wasn't necessary). Informally speaking, we meant the following: in typical GAN learning (e.g. Goodfellow's original formulation) the discriminator is able to say "in this region of space A, the probability of a sample being fake is x%". Which provides the generator with the information of how well it does in said region. However, the discriminator has usually no way of telling the generator "you should move probability mass over to this region B which is far, far away from A, because there is a lack of generated density there". Thus, the discriminator cannot tell the generator how to globally move its mass (it just gets local gradient information at the points where it currently generates). In particular, the generator cannot move samples across regions where the real world data has no support. As soon as generator samples appear at the border of such regions, they are penalized and move back to regions with real world support where they come from. Moving again means that samples "a" are associated with random variables "z", and small changes in the generator lead to small changes of "a" ("z" is mapped to a slightly different "a"). Thus, it is impossible to move samples from one island of real world support to another island of real world support.

---

### Public Comment · ~Leon_Boellmann1 · 2017-11-04
**Fundamentally the same with Moment Matching Networks**

This paper is interesting. It relates the GAN learning game to potential field. It seems to me it is fundamentally the same as Moment Matching Networks, in the sense that it aims to minimize the distance between two distributions. In particular, I have the following questions:

1. The discriminator aims to learn the potential function Phi. The optimal D(.) = Phi(.). Since we already know the expression of Phi, why don't we just use Phi for the generator optimization in eq (21)?

2. If the answer to question (1) is affirmative. Then the formulation is exactly the same as "Generative Moment Matching Networks", where the generator is updated to minimize some distance between two distributions. The only difference is that in GMM networks, Phi(.) is in the form of kernel, while in this paper Phi(.) is in the form of potential function given by eq. (1).

3. Theorem 2 and the title shows that the proposed GAN converges to the optimal Nash equilibrium. This holds in the condition that the objective functions can reach global minimum. For the original GAN proposed by Ian Goodfellow, if the optimization is with respective to D(.) and pg(.) in the function space, the Nash equilibrium is also unique!

Generally, one of the main challenges of GAN is that it is not convex-concave in the network parameters. If it is formulated as optimization in terms of the function space D(.) and pg(.), the local optimum is exactly the global one with D(.)=1/2 and pg=pd. The global nash equilibrium of Coulomb GAN is important, but is not a major contributions, considering that all other GANs have a unique Nash equilibrum in D(.) and pg(.). The major contribution of this paper is to provide a new perspective of formulating GAN, which will enable us to borrow the ideas from other fields to design systematic training methods or innovative GAN structures.

---

> ### Author Response · Authors · 2017-11-06
> **Coulomb GANs have no Local Nash Equlibria**
>
> Hi, thank you for your comment. You are right, as we discussed within the paper, MMD is indeed the closest existing work, and the two differences you pointed out are exactly the two fundamental key novelties of our approach, which allow us to make theoretical guarantees that go over what MMD claim.  To your specific questions:
>
> 1. Calculating the true Phi(...) would involve calculating the pairwise interactions between all pairs of training set data points, which is not feasible. Averaging over many \hat{Phi} (i.e., the potential function spanned by only a single mini-batch), is also very difficult:  Even averaging over many mini-batches is infeasible because of the high variance in approximating the whole field when using only a subset of samples. The key idea is to "store" the average field in the discriminator which includes also a smoothing effect and tracks the change of the field. Since the field created by the previous generator is close to the actual field, the discriminator can track the current field. We would expect that MMD approaches would also benefit greatly from adapting this approach. To the best of our knowledge, no MMD paper has ever made this connection and proposed this solution to the problem (from which MMDs do clearly suffer).
>
> 2.  Yes, you are right, our kernel is very different from the Gaussian Kernel that is used in MMD. In fact, as we show in Theorem 1,  our kernel can guarantee that we learn the correct distribution (if points are freely movable). Moreover, Hochreiter & Obermayer (2001) show that Gaussian Kernels *do not* guarantee convergence to a unique solution. So our choice of kernel is a very crucial improvement over an MMD approach.
>
> 3. This is a tricky point, and we might not have explained this well in the paper: Yes, the original GAN by Goodfellow has a unique Nash Equilibrium when the two distributions match perfectly. This is a true Nash Equlibrium according to the original definition: neither D or G can improve on their own strategy. However, due to the way we train GANs (using gradient based methods), there are also many local Nash Equilibria: situations where neither D or G can improve their own strategy within their local environments. They have to follow their gradients and cannot "jump" out of this local environment. We describe an example of this in appendix A1 of the paper (Mode collapse is a special case of such a local optimum). These are not Nash Equilibria in a global sense as assumed in the original definition, as better strategies for both players exist; but those strategies are unreachable with gradient based methods.
>
> To put it another way: What we ultimately want is to match two distributions. The optimization problem created by Goodfellow's GAN is littered with many local Nash Equilibria (where the distributions don't match but gradients vanish) where optimization will get stuck, even though we know that there is a global NE (where the distributions match) somewhere else. The Coulomb GAN's error surface does not contain such local Nash Equilibria. You cannot construct a situation where a Coulomb GAN's gradients vanish unless you are at the unique global optimum.

---

> > ### Public Comment · ~Leon_Boellmann1 · 2017-11-06
> > **Thanks a lot!**
> >
> > Thanks a lot! I think it is a very good paper. I hope it will be accepted.

---

> > > ### Public Comment · ~Leon_Boellmann1 · 2017-11-08
> > > **Follow up question on local NE**
> > >
> > > Dear authors,
> > > I think about the argument of the local NEs and still have a couple questions. The generator objective function is to minimize f = - \sum_j log(D(G(z_j))). We take derivative of this w.r.t. the network parameters theta_g, by chain rule we have
> > > df/d theta =  - \sum_j (1/D(G(z_j))) * (d D(x) / dx |x = G(z_j))  *(dG /d theta).
> > >
> > > For traditional GANs, (d D(x) / dx |x = G(z_j))  = 0, which yields the gradients equal to zero. I think Coulomb GAN makes  (d D(x) / dx |x = G(z_j)) non-zero whenever pg is not equal to pd. However, the third term (dG /d theta) could still be zero. We recently came up with a new training algorithm, which exactly faces the same problem.
> > >
> > > According to my understanding, the proposed GAN considerably reduces the local NEs but not remove all local NEs, because this problem is fundamentally a nonconvex problem. Please let me know if my understanding is correct.

---

> > > > ### Author Response · Authors · 2017-11-10
> > > > **Clarification**
> > > >
> > > > Hi! Interesting to hear that you're facing a similar argument! First off: Note that in Coulomb GANs, the generator objective does not include a log; but that's just a minor note.
> > > >
> > > > If I get you right, you're saying that the Generator might get stuck. This is indeed true for generators that haven't got enough capacity to move points freely along the field generated by the potential (e.g. a super small generator, this is why we have Assumption A1 in Theorem 2). But as long as the generator can still move it's points, Theorem 1 guarantees that there are no local NE.
> > > >
> > > > But I'm not sure I understood you correctly: another interpretation of your question is that d_G / d_Theta = 0 for all possible z_i, but that's only possible if G is a constant function. Is that what you meant?

---

### Public Comment · ~Emanuele_Sansone1 · 2017-11-27
**Proof of main theorem 1**

This paper represents the first attempt to formulate GANs as a minimization problem rather than a standard mini-max.

The overall idea is interesting, but I have some concerns regarding the proof of convergence to the global Nash equilibrium (proof of the main Theorem 1):
1. The authors study the function \nabla^2k(a,b) and claim that its minimum corresponds to Eq. 16. The methodology to check the minimality of the function (Eqs. 18-19) is not conventional. One should compute the Hessian matrix and then see if it is positive (semi-)definite.
2. The last inequality in Eq. 21 is not valid in general. The authors should specify the conditions of validity for that. Note that the bound of \nabla^2k(a,b) is obtained by setting \epsilon to zero. One can find non-zero values of  \epsilon for which the equality does not hold.

---

> ### Author Response · Authors · 2017-11-29
> **Notes about the proof**
>
> Hi! Thanks for your thoughtful comments, we also think Coulomb GANs are an interesting new avenue. For your questions:
>
> ad 1:The proof that Eq (16) is a minimum is formally correct: Since the derivative in Eq (18) is != 0 everywhere, the only extreme points we need to check are the boundary. Here, the minimum is at r = 0, as the function increases with r (see Eq 18).
>
> ad 2: In order to see the general validity of (21), set epsilon to 0. From (19) follows that the expression is continuously increasing with increasing epsilon. Therefore like with r there is a minimum at the boundary point of \epsilon=0.

---

### Decision · Program_Chairs · 2018-01-29
**ICLR 2018 Conference Acceptance Decision**

**Decision:**

Accept (Poster)

**Comment:**

The paper provides an interesting take on GAN training based on Coulomb dynamics. The proposed formulation is theoretically well motivated and authors provide guarantees for convergence. Reviewers agree that the theoretical analysis is interesting but are not completely impressed by the results. The method addresses mode collapse issue but still lacks in sample quality. Nevertheless, reviewers agree that this is a good step towards the understanding of GAN training.